# Iron Oxide–Silica Core–Shell Nanoparticles Functionalized with Essential Oils for Antimicrobial Therapies

**DOI:** 10.3390/antibiotics10091138

**Published:** 2021-09-21

**Authors:** Cristina Chircov, Maria-Florentina Matei, Ionela Andreea Neacșu, Bogdan Stefan Vasile, Ovidiu-Cristian Oprea, Alexa-Maria Croitoru, Roxana-Doina Trușcă, Ecaterina Andronescu, Ionuț Sorescu, Florica Bărbuceanu

**Affiliations:** 1Department of Science and Engineering of Oxide Materials and Nanomaterials, University Politehnica of Bucharest, 011061 Bucharest, Romania; cristina.chircov@yahoo.com (C.C.); neacsu.a.ionela@gmail.com (I.A.N.); bogdan.vasile@upb.ro (B.S.V.); alexa_maria.croitoru@upb.ro (A.-M.C.); 2National Research Center for Micro and Nanomaterials, University Politehnica of Bucharest, 060042 Bucharest, Romania; truscaroxana@yahoo.com; 3Faculty of Medical Engineering, University Politehnica of Bucharest, 011061 Bucharest, Romania; mateimariaflorentina98@yahoo.com; 4Department of Inorganic Chemistry, Physical Chemistry and Electrochemistry, University Politehnica of Bucharest, 1-7 Polizu St., 011061 Bucharest, Romania; ovidiu.oprea@upb.ro; 5Academy of Romanian Scientists, 54 Spl. Independentei, 050045 Bucharest, Romania; 6Institute for Diagnosis and Animal Health, 050557 Bucharest, Romania; ionut.sorescu@idah.ro (I.S.); florica.barbuceanu@idah.ro (F.B.); 7Faculty of Veterinary Medicine, University of Agricultural Science and Veterinary Medicine, 105 Splaiul Independentei, 050097 Bucharest, Romania

**Keywords:** magnetite nanoparticles, silica, core–shell nanoparticles, natural bioactive compounds, essential oils, antimicrobial therapy

## Abstract

Recent years have witnessed a tremendous interest in the use of essential oils in biomedical applications due to their intrinsic antimicrobial, antioxidant, and anticancer properties. However, their low aqueous solubility and high volatility compromise their maximum potential, thus requiring the development of efficient supports for their delivery. Hence, this manuscript focuses on developing nanostructured systems based on Fe_3_O_4_@SiO_2_ core–shell nanoparticles and three different types of essential oils, i.e., thyme, rosemary, and basil, to overcome these limitations. Specifically, this work represents a comparative study between co-precipitation and microwave-assisted hydrothermal methods for the synthesis of Fe_3_O_4_@SiO_2_ core–shell nanoparticles. All magnetic samples were characterized by X-ray diffraction (XRD), gas chromatography-mass spectrometry (GC-MS), Fourier-transform infrared spectroscopy (FTIR), dynamic light scattering (DLS), zeta potential, scanning electron microscopy (SEM), transmission electron microscopy (TEM), thermogravimetry and differential scanning calorimetry (TG-DSC), and vibrating sample magnetometry (VSM) to study the impact of the synthesis method on the nanoparticle formation and properties, in terms of crystallinity, purity, size, morphology, stability, and magnetization. Moreover, the antimicrobial properties of the synthesized nanocomposites were assessed through in vitro tests on *Staphylococcus aureus*, *Pseudomonas aeruginosa*, *Escherichia coli*, and *Candida albicans*. In this manner, this study demonstrated the efficiency of the core–shell nanostructured systems as potential applications in antimicrobial therapies.

## 1. Introduction

Human existence is essentially dependent on the action of microorganisms, as they play fundamental roles in the fixation of nitrogen, production of vitamins, photosynthesis, and decomposition of organic matter [1,2,3]. Nonetheless, a shift of the delicate balance between the immune system and microorganisms in favor of the latter could cause severe immunodeficiencies [2,4]. Accounting for millions of deaths each year worldwide, infectious diseases, which are caused by pathogenic bacteria, viruses, fungi, or parasites and can be directly or indirectly transmitted through air, water, food, or living vectors, have become a challenging threat to public health and a top priority area for health policy [2,5,6,7].

While ancient times witnessed the deaths of more than half of born individuals before reaching sexual maturity [8], the discovery of antimicrobial drugs has greatly impacted the global health system by significantly reducing morbidity and mortality associated with infectious diseases [9]. However, microorganisms have acquired or developed numerous resistance mechanisms over time against all commercially available antimicrobial drugs [9,10]. The main causes involved in the development of such mechanisms include genetic modifications of microorganisms, gratuitous over-prescription of antibiotics and broad-spectrum antibiotics, easy access to over-the-counter antibiotics, and contaminations due to antimicrobial drug manufacturing [11]. As antimicrobial resistance has led to ineffectiveness of currently existing drugs, a worldwide catastrophic issue for the public system, there is an urgent necessity to identify novel alternative strategies [5,7,9,11,12,13,14,15].

One of the most intensively studied alternative involves the use of natural compounds, either alone or in combination with conventional antimicrobial drugs to enhance their activity [7,13,16]. Studies have demonstrated the efficiency of various natural compounds, including plant-derived (e.g., essential oils (EOs), polyphenols), animal-derived (e.g., lactoferrin, lysozyme, lactoperoxidase), algal extracts, and microbial metabolites, which act through a series of mechanisms, such as microbial cell membrane rupture, nucleic acid process impairment, proton motive force decay, or adenosine triphosphate depletion [16,17,18].

Increasing scientific interest has focused on EOs, which are secondary plant metabolites consisting of complex mixtures of odoriferous and volatile organic components produced by different plant parts, including flowers, peels, seeds, leaves, buds, twigs, roots, or fruits [19,20,21,22]. Since early times, EOs have been widely used in numerous medical applications, including cosmetics, dermatology, aromatherapy, and self-care medical products, or for food preservation purposes [19,21]. Currently, they are widely found in commercial applications, such as medicines, and there are many clinical trials performed [23]. There are many research studies demonstrating their biological benefits, including antioxidant, anti-inflammatory, anti-cancer, antimicrobial (antibacterial, antiviral, antifungal, and antiparasitic), analgesic, sedative, and wound healing [19,20,21]. With regard to the current antimicrobial resistance crisis, EOs have proven their efficiency by targeting microbial cell walls or membranes, cellular respiration processes, or quorum sensing mechanisms [19,24]. Nevertheless, despite their tremendous potential, EOs have a series of limitations associated with their increased hydrophobicity, volatility, lipophilicity, and oxidation susceptibility and decreased solubility and stability [21,25]. Therefore, in order to inhibit the development of microbial pathogens, their combination with nanotechnology-based approaches is necessary [21]. In this manner, nanocarriers could protect them against thermal and photodegradation, while controlling their release and increasing their solubility, bioavailability, bioaccesibility, and concentration at the target site [18,25,26,27,28]. Moreover, since nanoparticles have shown a high potential towards inactivating various microbial species through intrinsic antimicrobial features, as they are capable of microbial membrane and wall permeation and disruption, reactive oxygen species generation, and intracellular component damaging, systems comprising nanoparticles functionalized with EOs could offer synergistic effects [14,29,30,31]. Specifically, magnetite nanoparticles are well-known for their antimicrobial properties, both for their intrinsic properties that can be potentiated by hyperthermia effects and as antimicrobial drug carriers [32]. Furthermore, owing to its high surface reactivity due to the high amount of hydroxyl groups onto the surface, silica was added in order to enhance the immobilization of EOs within the systems (Figure 1).

In this context, the current study involves synthesizing composite core–shell nanoparticles comprising magnetite cores and silica layers obtained through two different methods, namely co-precipitation and the microwave-assisted hydrothermal method. The co-precipitation method was selected as one of the most commonly used techniques for the synthesis of magnetite nanoparticles due to its convenience, low cost, and possibility of large-scale production. However, as the obtained nanoparticles are generally characterized by decreased crystallinity and have a high tendency for agglomeration and oxidation, the microwave-assisted hydrothermal method was applied in order to overcome such limitations. In this manner, this method provides the advantages of synthesizing nanoparticles with a narrow size and shape distribution, due to homogenous high temperature and pressure conditions [21,33]. After the synthesis process, the nanosystems were functionalized with three different EOs, namely thyme, rosemary, and basil.

## 2. Results and Discussion

The Fe_3_O_4_@SiO_2_ nanosystems were obtained through two different methods, namely co-precipitation (CP) and the microwave-assisted hydrothermal technique using the Synthwave (SW) equipment, and functionalized with three types of EOs, i.e., thyme, rosemary, and basil. The eight samples obtained and their associated labels are summarized in Table 1.

On the one hand, the present study investigated the influence of the synthesis method on the outcome properties of the Fe_3_O_4_@SiO_2_ in terms of mineral phase formation, crystallinity, morphology, size, surface charge, magnetic properties, and thermal stability. On the other hand, after synthesis, three different EOs, namely thyme, rosemary, and basil, were immobilized onto the surface of the nanosystems, and their antimicrobial properties were assessed through in vitro assays. In contrast to other research studies focused on the synthesis of Fe_3_O_4_@SiO_2_ core–shell nanoparticles [34,35,36], this study did not involve the use of surfactants or porogen agents.

The core–shell structure of the composite nanosystems was demonstrated through Bright Field Transmission Electron Microscopy (BF-TEM) images (Figure 2), which show clusters of 1 to 10 nanoparticles surrounded by a thick layer of silica. Subsequently, TEM images were used to determine the core nanoparticle size and the shell layer thickness. Specifically, 150 cores and 100 shell regions were measured using the ImageJ software and based on the obtained information, corresponding size distributions were created and fitted using the Gaussian curve fit available in the Origin software (Figure 3). Considering the FWHM of the fit, the average magnetite nanoparticle size was 6.7 nm, while the average thickness of the layer, 9.5 nm. Additionally, the nanosystems were characterized by an increased size uniformity, which is in accordance to similar nanosystems obtained through the microemulsion technique [37]. Since the primary advantage of the microemulsion synthesis is represented by the precise control of nanoparticle size through the adjustment of water, oil, and surfactant present within the system, these results prove the efficiency of the present study to overcome current limitations associated with co-precipitation and hydrothermal techniques [33,38].

Moreover, TEM images demonstrate the spherical shape of the magnetite nanoparticles. In this context, previous studies reported a quasi-cubic shape of the nanoparticles, which was maintained after coating with silica, which was associated with increased chemical stability provided by the silica shell [39]. Therefore, it could be safe to assume that the present Fe_3_O_4_@SiO_2_ core–shell nanoparticles are characterized by a prolonged shelf-life due to the presence of the SiO_2_ component.

The crystallinity and mineral phase formation of the Fe_3_O_4_@SiO_2_ nanoparticles were assessed through X-ray Diffraction (XRD) (Figure 4). Both diffractograms show the formation of a single crystalline phase, demonstrated through the diffraction peaks characteristic for magnetite (Fe_3_O_4_) in the Fd-3m cubic crystal system and the associated Miller indices (according to the PDF 00-065-0731 [40]). Additionally, the presence of the amorphous halo at the 2θ angles of 20–25° can be attributed to the silica layer within the nanosystems [41,42].

Furthermore, by comparing the two samples, it can be said that the microwave-assisted hydrothermal method is associated with a higher percentage of silica within the sample, as the diffraction halo within the Fe_3_O_4_@SiO_2__SW sample has a higher intensity. Consequently, the crystallinity of this sample is reduced, as the intensity of the diffraction peaks for sample Fe_3_O_4_@SiO_2__CP is slightly increased.

The crystallite size for both samples was calculated using the Debye–Scherrer equation:(1)$$\mathrm{FWHM}=\frac{K\lambda }{\mathrm{Dcos}\theta },$$
where FWHM is the full width at the half peak height, K is the Scherrer constant that varies between 0.89 and 0.94, λ is the X-ray wavelength, D is the crystallite size, and θ is the diffraction angle [43,44].

In this manner, the average crystallite size calculated as the mean value between all peaks is presented in Table 2. As expected, the crystallite size for the nanoparticles obtained through the hydrothermal method is higher due to the crystal growth under increased pressure and temperature conditions. Moreover, the crystallite size of the nanosystems is considerably lower than other results available in the scientific literature [45].

High Resolution-TEM (HR-TEM) and Selected Area Electron Diffraction (SAED) confirmed the formation of magnetite nanoparticles as a single crystalline phase (Figure 5). As it can be observed, the identified Miller indices measured from the SAED rings matched the ones within the XRD diffractograms. Moreover, the presence of amorphous silica leads to the formation of the halo, which can be seen between the central spot and the first diffraction ring.

Subsequently, the Fe_3_O_4_@SiO_2_ core–shell nanosystems were subjected to elemental mapping in order to determine the nature of the present elements and to further confirm the formation of the iron oxide core and the silica shell. Thus, Figure 6 depicts the identified elements and their distribution within the composite nanosystems. Specifically, it can be seen that Fe is exclusively found within the magnetic cores, while Si is found throughout the system, as components of the silica layer. Therefore, both the nature of the constituents and the core–shell structure of the nanosystems were confirmed.

The Gas Chromatography-Mass Spectrometry (GC-MS) chromatograms revealed the presence of a series of volatile compounds characteristic for the EOs that were used for the experiment. The compounds identified within the EO-functionalized nanosystems and their retention time are summarized in Table 3. The chromatogram profiles for each EO and EO-functionalized core–shell nanoparticles and the mass spectra for each of the major compounds identified within the EOs can be found in the Appendix A. Additionally, the compounds identified within EOs and their retention time have also been added to the Appendix A.

The identified compounds are consistent with previously published studies investigating the compositional concentration of thyme, rosemary, and basil EOs [46,47,48,49]. It can be seen that the major compound identified within the thyme EO-functionalized samples is thymol and p-cymen-7-ol. Thymol is well-known for its antimicrobial activities and low MIC values. The peak present within the CP sample is considerably higher, which could be attributed to a higher concentration and, consequently, to an increased antimicrobial activity. The rosemary EO-functionalized samples presented peaks for eucalyptol and (+)-2-bornanone as major compounds. For the basil EO-functionalized systems, the major compounds identified are trans-linalyl formate and estragole.

Fourier Transform Infrared (FT-IR) Spectroscopy was used for the assessment of the functional groups present within the synthesized samples. In this context, Figure 7a presents the FT-IR spectra registered for all eight samples, namely for the simple Fe_3_O_4_@SiO_2__CP and Fe_3_O_4_@SiO_2__SW samples, and for the thyme, rosemary, or basil EO-functionalized Fe_3_O_4_@SiO_2__CP and Fe_3_O_4_@SiO_2__SW samples. There were five absorption bands at 445, 559, 795, 952, and 1066 cm^−1^ registered within all eight samples. The sharp bands at 445 and 1066 cm^−1^ correspond to the Si–O–Si or O–Si–O bending mode, while the band at 795 cm^−1^ is assigned to the Si–O–Si symmetric stretch. The Fe–O stretching mode characteristic for Fe_3_O_4_ is shown at 559 cm^−1^ [42,50]. The absorption peak at 952 cm^−1^ is attributed to the Fe-O-Si stretching vibration, thus demonstrating the formation of the silica layers onto the iron oxide core [51]. The wide absorption band at 3371 cm^−1^ present in all samples corresponds to the O-H stretching mode [42,50].

The square marked in the FT-IR spectra represents the wavenumber region where the presence of EOs can be observed. Specifically, Figure 7b–d presents the absorption peaks found in the functionalized samples that are characteristic to the EO that was used. While all samples were successfully functionalized, the absorption bands within the samples synthesized through the hydrothermal method have higher intensities, which could be related to higher functionalization yields. The wavenumbers for each absorption band and the associated bonds [52,53,54] are summarized in Table 4.

Considering the chemical structures of the main components found within thyme, rosemary, and basil EOs, the absorption peaks registered demonstrate immobilization of the EOs onto the core–shell nanoparticles. Specifically, considering thymol, eucalyptol, and estragole as the primary compounds of thyme, rosemary, and basil EOs, respectively, their characteristic reactive groups, i.e., O-H (phenol), O-H (alcohol), and -C-O, were found on the FT-IR spectra. As the mentioned compounds are well-known for their antimicrobial activity, it could be safe to assume that the antimicrobial activity of the EOs-functionalized core–shell nanoparticles should be ensured. However, since the rosemary EO-functionalized samples lack key absorption peaks specific to rosemary EO, such as C-O and C=O specific for camphor, and the peaks present have significantly low intensities, it could be concluded that the loading efficiency was reduced in this case. For better visualization, the chemical structures of thymol, eucalyptol, and estragole were represented in Figure 8.

Scanning Electron Microscopy (SEM) analysis allowed for visualization of the composite nanosystems morphology (Figure 9). In this context, both Fe_3_O_4_@SiO_2__CP and Fe_3_O_4_@SiO_2__SW samples exhibited a quasispherical shape with a considerable tendency for agglomeration. It can be seen that the composite systems are characterized by nanoscaled dimensions. Larger sizes were registered for the nanoparticles obtained through the hydrothermal method, thus confirming the previous results. Moreover, there are several nanoparticle aggregates with dimensions in the range of 200 nm corresponding to the Fe_3_O_4_@SiO_2__SW sample. By contrast to other studies investigating the formation of Fe_3_O_4_@SiO_2_ core–shell nanoparticles which report considerably larger core sizes and shell thicknesses [45,55,56], these results demonstrate the formation of nanoscaled core–shell systems with high uniformity and reduced sizes.

The hydrodynamic diameter and zeta potential of the core–shell nanoparticles were assessed both before and after functionalization with EOs. Five measurements were performed on each sample, and the mean values were calculated accordingly (Table 5).

Figure 10 presents a visualization of the mean hydrodynamic diameter and zeta potential values registered for the eight samples. The highest hydrodynamic diameter values recorded are associated with the simple Fe_3_O_4_@SiO_2_ samples. Since the hydrodynamic diameter is defined as the diameter of the hypothetical solid sphere formed through the attachment of solvent molecules onto the surface of the core nanoparticles [57,58], this can be explained by the high number of interactions between the surface hydroxyl groups characteristic to silica and water molecules. Therefore, since EOs are attached to the composite nanosystems through binding to the reactive groups present onto the surface, interaction with the solvent will be limited. Consequently, the hydrodynamic diameter will be reduced. Moreover, the previous results are confirmed by the higher values registered for the nanosystems synthesized through the hydrothermal method. The size distribution for each sample and the associated correlation graph can be found in the Appendix A.

The zeta potential is generally considered a reference of the stability of the nanosystems dispersed into a solvent, as it reflects the amount of surface charges. Specifically, at values close to 0 (−25 mV to 25 mV), nanoparticles have a tendency to form aggregates due to attractive forces present at the surface. Therefore, zeta potential values lower than −25 mV and higher than 25 mV are associated with stable dispersions [59,60,61]. As it can be seen in Figure 4, the lowest zeta potential values were recorded for samples Fe_3_O_4_@SiO_2_@thyme_CP and Fe_3_O_4_@SiO_2_@rosemary_CP, which had the smallest hydrodynamic diameters. By contrast, the values for the nanosystems obtained through the hydrothermal technique were in the range of −24–−18 mV, demonstrating a higher agglomeration tendency due to higher surface reactivity specific to the synthesis method.

The thermal behavior of the core–shell nanoparticles was assessed through Thermogravimetry and Differential Scanning Calorimetry (TG-DSC). In this context, Figure 11 presents the TG-DSC curves for the Fe_3_O_4_@SiO_2__CP and Fe_3_O_4_@SiO_2__SW nanosystems. In the temperature range of 20–180 °C, there is a mass loss of 4.69% and 5.77% accompanied by an endothermic effect with a minimum at 66.4 °C and 66.1 °C, respectively. The higher mass loss for the system obtained through the hydrothermal method could be associated with a higher amount of water molecules absorbed and adsorbed within the nanoparticles. In this manner, the higher affinity for water adsorption could be attributed to the higher hydrodynamic diameter previously reported. Moreover, the absence of the exothermic effects at ~300 °C and ~600 °C that are generally attributed to the transformation of magnetite into maghemite and subsequently of maghemite to hematite demonstrate that the iron oxide core is protected from thermal oxidation [62,63]. This hypothesis was also confirmed after the analysis, as the residual powder maintained its magnetic behavior.

Subsequently, samples Fe_3_O_4_@SiO_2_@thyme_CP, Fe_3_O_4_@SiO_2_@rosemary_CP, Fe_3_O_4_@SiO_2_@basil_CP, Fe_3_O_4_@SiO_2_@thyme_SW, Fe_3_O_4_@SiO_2_@rosemary_SW, and Fe_3_O_4_@SiO_2_@basil_SW were subjected to the same thermal treatment in order to assess the loading efficiency of each sample (Figure 12). All samples were characterized by an initial mass loss in the temperature range 20–180 °C through an endothermic process with a minimum at 60–70 °C due to the elimination of volatile compounds present within the EOs and residual solvent molecules. The second mass loss was registered between 180–500 °C associated with an exothermic effect with the maximum at 300–400 °C, indicating the oxidation of less volatile compounds present within the EOs and the elimination of hydroxyl groups from the surface of the nanoparticles. The final mass loss occurs between 500–900 °C. Table 6 summarizes the mass losses mentioned for each sample and the associated thermal effects.

As it can be seen in Table 6, the mass losses associated with the nanosystems obtained through the hydrothermal method are considerably higher in the temperature interval 20–180 °C. As this interval is attributed to the elimination of most of the EO compounds, it could be stated that these nanosystems are characterized by a higher loading capacity possibly due to a higher surface reactivity, as shown by the zeta potential measurements. Moreover, the estimated loading efficiency for each system, calculated as the difference between the mass loss associated with the EO-functionalized core–shell nanoparticles and the mass loss associated with the unfunctionalized core–shell nanoparticles, confirms this hypothesis. Additionally, the highest loading efficiency is attributed to thyme, followed by basil and, subsequently, rosemary. This trend is respected in both types of synthesis methods. These results are consistent with the information obtained from the FT-IR spectra.

The magnetic properties of the Fe_3_O_4_@SiO_2_ core–shell nanoparticles were measured through the Vibrating Sample Magnetometry analysis (Figure 13). The superparamagnetic behavior of the nanoparticles is demonstrated by the S-shaped hysteresis curve of magnetization versus applied magnetic field with a width of zero [64,65,66]. Such behavior further allows for developing drug delivery applications, as the drug-carrying nanoparticles are magnetized under an external magnetic field and lose their magnetization once the magnetic field is removed [67,68,69]. Furthermore, the values associated with the saturation magnetization (M_s_), remanence magnetization (M_r_), and coercivity field (H_c_) of the core–shell nanoparticles are shown in Table 7.

Considering the silica shell thickness of 9.5 nm incorporating 3 magnetic cores of about 6.7 nm, as given by TEM results, and the density for silica and magnetite of 2.65 g/cm^3^ and 5.18 g/cm^3^ [21], respectively, it can be deduced that the percentage of magnetite mass within the composite nanosystem is about 31%. Therefore, since the M_s_ of the core–shell nanosystems is 20.32 emu/g and 16.95 emu/g, the M_s_ of the magnetic core would be approximately 65 emu/g and 55 emu/g, respectively. Moreover, besides normalizing the magnetization by sample mass, its decrease could also be explained by the diamagnetic behavior of the silica shell [41,70]. While the magnitude of the demagnetizing field generated in the opposite direction of the applied external field is generally small for diamagnetic materials [71], it cannot be neglected in this case, as a high field (10,000 Oe) is applied [41,72].

Furthermore, low M_r_ and H_c_ values are specific to small nanoparticles with superparamagnetic behavior [73]. In this context, as previous results proved larger sizes for the nanosystems obtained through the hydrothermal method, the higher M_r_ and H_c_ values for the Fe_3_O_4_@SiO_2__SW were expected.

In this manner, the potential of the core–shell nanosystems for the use in hyperthermia-associated controlled release of bioactive substances was confirmed.

In the context of antimicrobial studies, the absence of interferences due to initial bacterial and fungal contamination of the samples was confirmed, as the nanoparticles seeded at the concentration of 1 mg/mL and incubated for 14 days at 37 °C and 28 °C, respectively, showed no signs of bacterial strains or yeast development.

Subsequently, the concentrations of 1, 2, and 4 mg/mL and 0.1, 0.2, and 0.4 µL/mL of Fe_3_O_4_@SiO_2__CP, Fe_3_O_4_@SiO_2_@thyme_CP, Fe_3_O_4_@SiO_2_@rosemary_CP, Fe_3_O_4_@SiO_2_@basil_CP, Fe_3_O_4_@SiO_2__SW, Fe_3_O_4_@SiO_2_@thyme_SW, Fe_3_O_4_@SiO_2_@rosemary_SW, and Fe_3_O_4_@SiO_2_@basil_SW and of thyme, rosemary, and basil EOs, respectively, were tested against *Staphylococcus aureus* ATCC 25923 (2 × 10^6^ UFC/mL test), *Pseudomonas aeruginosa* ATCC 27853 (4 × 10^5^ UFC/mL test), *Escherichia coli* ATCC 25922 (3.6 × 10^6^ UFC/mL test), and *Candida albicans* ATCC 10231 (1 × 10^5^ UFC/mL test).

The results are summarized in Table 8. Briefly, the concentrations of 1 mg/mL and 0.1 µL/mL could not inhibit any of the microbial cultures. The concentration of 4 mg/mL of Fe_3_O_4_@SiO_2_@thyme_CP and Fe_3_O_4_@SiO_2_@thyme_SW samples inhibited the development of *Staphylococcus aureus*, *Escherichia coli*, and *Candida albicans* cultures. However, the concentration of 0.4 µL/mL only inhibited the development of *Staphylococcus aureus*. In this manner, the contribution of the Fe_3_O_4_@SiO_2_ substrate to the antimicrobial activity of the nanosystem, but not alone or in combination with the other two types of EOs, was demonstrated. Additionally, immobilizing the EOs onto the surface of the nanosystems prevents their volatilization, thus ensuring the antimicrobial activity of their components. *Pseudomonas aeruginosa* culture development was not inhibited by these concentrations. The concentration of 2 mg/mL of samples Fe_3_O_4_@SiO_2_@thyme_CP and Fe_3_O_4_@SiO_2_@thyme_SW, but not 0.2 µL/mL of thyme EO, inhibited the development of *Staphylococcus aureus.* Thus, the minimum inhibitory concentration (MIC) value against the *Staphylococcus aureus* strain of the two nanoparticle samples was established at 2 mg/mL and 0.4 µL/mL for the thyme EO. Furthermore, only sample Fe_3_O_4_@SiO_2_@thyme_CP inhibited the development of *Escherichia coli* and *Candida albicans* cultures. The MIC value against the two microbial species was determined at 2 mg/mL for sample Fe_3_O_4_@SiO_2_@thyme_CP and 4 mg/mL for sample Fe_3_O_4_@SiO_2_@thyme_SW.

This outcome is in accordance with previous results, as GC-MS, DSC-TG, and FT-IR analyses indicated a considerably higher loading efficiency for the thyme EO-functionalized core–shell nanoparticles. Furthermore, previous studies investigating the antimicrobial efficiency of various EOs and their major compounds reported a significantly higher activity for eucalyptol, the major rosemary EO compound, compared to thymol, the major thyme EO compound [74]. Other results report considerably high MIC values for estragole, the major compound of basil EO [75]. Additionally, the higher antibacterial activity of thymol against Gram positive than Gram negative bacterial strains is well-known and reported in the literature [76,77,78], which explains the higher antibacterial activity against *Staphylococcus aureus*. Precise MIC values for each of the three compounds, as reported in the literature, are summarized in Table 9.

Moreover, the increased antimicrobial activity of the nanoparticles synthesized through the hydrothermal method could be related to the smaller Fe_3_O_4_ nanoparticle size, which is generally associated with a higher capacity to penetrate and disrupt microbial cell walls [9]. Furthermore, since there was a higher amount of silica within these samples, the intrinsic antimicrobial activity of the Fe_3_O_4_ nanoparticles could be reduced due to the limited contact between the cores and the bacterial cells.

## 3. Materials and Methods

### 3.1. Materials

Ferrous sulfate heptahydrate (FeSO_4_·7H_2_O), ferric chloride hexahydrate (FeCl_3_·6H_2_O), ammonium hydroxide 25% (NH_4_OH), tetraethyl orthosilicate (TEOS), chloroform and ethanol (EtOH) were purchased from Sigma-Aldrich Merck (Darmstadt, Germany). Thyme, rosemary, and basil EOs were purchased from Carl Roth (Karlsruhe, Baden-Württemberg, Germany). All chemicals were of analytical purity and used with no further purification.

Gram-positive and Gram-negative bacterial and fungal species involved in numerous human pathologies due to their potential to generate microbial biofilms and to manifest antibiotic resistance were selected. Specifically, reference ATCC strains from the collection of the Institute for Diagnosis and Animal Health (I.D.A.H.), namely *Staphylococcus aureus* ATCC 25923, *Pseudomonas aeruginosa* ATCC 27853, *Escherichia coli* ATCC 25922, and *Candida albicans* ATCC 10231 were used for antimicrobial assays. For the cultivation and testing of bacterial strains, Oxoid BHI broth was used, while for *Candida albicans*, Oxoid Sabouraud Liquid Medium.

### 3.2. Methods

#### 3.2.1. Fe_3_O_4_ NPs Synthesis

Fe_3_O_4_ NPs were synthesized through two different methods, namely co-precipitation and microwave-assisted hydrothermal synthesis.

FeSO_4_·7H_2_O and FeCl_3_·6H_2_O were dissolved in deionized water in a 1:2 molar ratio. Using a peristaltic pump, the iron precursor solution was dripped into an NH_4_OH-containing alkaline solution that allowed for the co-precipitation of a black precipitate consisting of iron oxide nanoparticles. The nanoparticles were decanted using a high-power permanent magnet and rinsed with deionized water until a neutral pH.

The second method followed a similar procedure as the co-precipitation technique, except for the fact that the black precipitate was transferred into a polytetrafluoroethylene (Teflon) vial that was introduced into the Milestone Synthwave equipment. The microwave-assisted reaction was carried under 60 bar (N_2_) pressure, and the vial was irradiated with microwaves at 80 °C for 30 min and 10% stirring. Subsequently, the obtained nanoparticles were rinsed with deionized water until a neutral pH.

#### 3.2.2. Fe_3_O_4_@SiO_2_ Core–Shell Systems Synthesis

The silica layer was obtained through a conventional sol-gel method. Specifically, Fe_3_O_4_ NPs were redispersed in a EtOH:H_2_O solution of 7.7:1 molar ratio using an ultrasonication bath. Subsequently, NH_4_OH and TEOS (as the silica precursor) were added to the nanoparticle dispersion in order to obtain the core–shell systems consisting of Fe_3_O_4_@SiO_2_ at a 1:6 weight ratio. The obtained mixture was kept under continuous magnetic stirring for 24 h at room temperature. Furthermore, the nanosystems were decanted using a high-power permanent magnet and rinsed with deionized water until neutral pH. The nanosystems were dried at 80 °C overnight.

#### 3.2.3. Fe_3_O_4_@SiO_2_@EOs Systems Synthesis

Firstly, 100 μL from each EO (i.e., thyme, rosemary, and basil EOs) was added to 1 g of Fe_3_O_4_@SiO_2_ NPs from each synthesis method. Thus, 8 samples were obtained and labeled according to the synthesis method, i.e., co-precipitation (CP) and microwave-assisted hydrothermal method (MH), and the type of EO used for functionalization.

#### 3.2.4. Physicochemical Characterization

##### X-ray Diffraction (XRD)

Grazing incidence XRD (GIXRD) was carried out using a PANalytical Empyrean diffractometer (PANalytical, Almelo, The Netherlands), using the Cu K_α_ radiation (λ = 1.541874 Å) equipped with a hybrid monochromator 2×Ge (220) for Cu and a parallel plate collimator on the PIXcel3D detector. The XRD diffractograms were acquired in the range of 10–80° 2θ angles, with an incidence angle of 0.5°, a step size of 0.0256°, and the time for each step of 1 s. The crystallite size was determined using the Origin software.

##### Bright Field Transmission Electron Microscopy (BF-TEM). High-Resolution TEM (HR-TEM). Selected Area Electron Diffraction (SAED). Energy Dispersive X-ray Spectroscopy (EDXS)

A small amount of the samples was dispersed into deionized water, and after that 10 µL of the suspension was placed on a 400 mesh lacey carbon-coated copper grid at room temperature and analyzed using a High-Resolution 80–200 TITAN THEMIS transmission microscope equipped with an Image Corrector and 4 EDXS detector in the column, purchased from the FEI (Hillsboro, OR, USA). The microscope operates in transmission mode at a 200 kV voltage. The elemental mapping was made in STEM mode using a High Annular Angular Dark Field Detector (HAADF) and 4 EDXS detectors. Particle size distribution was assessed by creating histograms corresponding to the TEM images using the ImageJ software.

##### Gas Chromatography-Mass Spectrometry (GC-MS)

Qualitative analysis of the volatile and semi-volatile compounds from the vapor phase of the samples was performed through the extraction of the compounds from the surface of the nanoparticles and the EOs with a suitable solvent and the subsequent GC-MS analysis using an Agilent gas chromatograph coupled with a mass spectrometer Agilent Q-TOF, serial number US15173026/250146. Briefly, 10 mL of the extraction solvent (methanol) was added to 0.5 g of the EOs-functionalized nanoparticles and thoroughly mixed for 5 min. The mixture was then filtrated using 0.22 µm syringe filters and the extracts were diluted with methanol at a ratio of 1:1 (*v/v*). The EOs were diluted with methanol at a ratio of 1:1000 (*v/v*). The analytical separation conditions included: a GC-MS Agilent 7890B system comprising a gas chromatograph and a Q-TOF 7200 mass spectrometer, an Ultra Inert HP-5MS column with the length of 30 m, inner diameter of 0.25 µm, and stationary phase film thickness of 0.25 µm. The carrier gas was helium, with a flow of 1 mL/min. The temperature range involved 45 °C (3 min isotherm), followed by an increase of 4 °C/min until 200 °C (41 min isotherm), and a subsequent increase of 7 °C/min until 280 °C (63 min isotherm). The temperatures of the injector, transfer line, and at quadrupole were 280 °C, 250 °C, and 150 °C, respectively. Then, 1 µL was injected through the Pulsed splitless mode, splitless time of 0.3 min and purge of 100 mL/min for each sample. The parameters of the mass spectrometer included electron impact ionization at 70 eV, mass range of 41–850 uam, and an electron multiplier detector. Functional parameter verification was performed through auto-tunning before the working sequence. Data processing was performed with the MassHunter WorkStation software, and the spectra library used was NIST 2005 v.2.0 D (Agilent Technologies, Santa Clara, CA, USA). To avoid the appearance of false peaks originating from the extraction solvent, blank samples containing only the solvent were analyzed through the same conditions. There were no peaks within the chromatogram for the blank samples, demonstrating the purity of the solvent used for the desorption process.

##### Infrared (IR) Spectroscopy

IR spectra were obtained with a Thermo iN10-MX Fourier transform (FT)-IR microscope (Thermo Fischer Scientific, Waltham, MA, USA) with a liquid nitrogen-cooled mercury cadmium telluride detector with the measurement range between 4000–400 cm^−1^. Spectra collection was performed in reflection mode at a resolution of 4 cm^−1^. For each spectrum, 64 scans were co-added and converted to absorbance using the OmincPicta software (Thermo Scientific).

##### Scanning Electron Microscopy (SEM)

The morphology and size of the nanosystems were investigated by placing the samples in the analysis chamber of an Inspect F50 high-resolution microscope (Thermo Fisher—former FEI, Eindhoven, The Netherlands). The micrographs were acquired with a 30 KeV energy value at different magnifications.

##### Dynamic Light Scattering (DLS). Zeta Potential

The hydrodynamic diameter and the surface charge of the MNPs were obtained using the DLS technique (DelsaMax Pro, Backman Coulter, Brea, CA, USA). The systems were dispersed in deionized water (~6.9 pH) using an ultrasonication bath at the same concentrations (0.1 mg/mL). Five acquisitions were registered for each measurement.

##### Thermogravimetry and Differential Scanning Calorimetry (TG-DSC)

Thermogravimetric analysis was performed using an STA TG/DSC Netzsch Netzsch Jupiter 449 F3 equipment (Selb, Germany). The temperature range was between 20 and 900 °C in a dynamic atmosphere of 50 mL/min air with a heating rate of 10 K/min in an alumina crucible.

##### Vibrating Sample Magnetometry (VSM)

The magnetic properties of the Fe_3_O_4_@SiO_2_ nanoparticles were assessed through the VSM analysis (VSM, VersaLabTM 3T, Cryogen-free Vibrating Sample Magnetometer, Westerville, OH, USA). The magnetic field was applied between −10 and +10 kOe two times, with a step rate of 10 Oe/s. The magnetic behavior was studied at room temperature (25 °C).

#### 3.2.5. Antimicrobial Activity Assay

The protocols were performed according to previous antimicrobial studies [82]. Initially, the bacterial and fungal contamination of the samples was verified by inoculating them in BHI broth, anaerobic Oxoid broth, and Sabouraud Liquid at a concentration of 1 mg/mL and incubating for 14 days at 37 °C and 28 °C for bacterial and fungi media, respectively.

Attempts were made to establish the MIC of samples Fe_3_O_4_@SiO_2__CP, Fe_3_O_4_@SiO_2_@thyme_CP, Fe_3_O_4_@SiO_2_@rosemary_CP, Fe_3_O_4_@SiO_2_@basil_CP, Fe_3_O_4_@SiO_2__SW, Fe_3_O_4_@SiO_2_@thyme_SW, Fe_3_O_4_@SiO_2_@rosemary_SW, and Fe_3_O_4_@SiO_2_@basil_SW, but also of the thyme, rosemary, and basil EOs against the three bacterial strains and the *Candida albicans* strain by the dilution method in liquid culture medium (BHI broth for bacteria and Sabouraud Liquid for yeast).

Specifically, nanoparticle samples and thyme, rosemary, and basil EOs were each distributed at three different concentrations (1 mg/mL, 2 mg/mL, 4 mg/mL and 0.1 µL/mL, 0.2 µL/mL, 0.4 µL/mL, respectively) in tubes containing 1.25 mL liquid medium. Subsequently, 0.125 mL cultures of 18 h diluted with physiological solutions of MF Standard 0.5 (BioMerieux) for bacteria and MF 2 for *Candida albicans* were inoculated.

The number of colony-forming units (CFU)/mL inoculum culture and, consequently, the number of CFU/0.125 mL inoculated cultured and the number of CFU/1.375 mL test were determined. Negative controls for culture media and all 11 samples (8 nanoparticle samples and 3 EO samples) were used to certify the absence of interference from bacterial and fungal contaminants during testing, as well as positive controls for the development of bacterial strains and yeast.

Incubation was performed at a temperature of 37 °C for 72 h, and the development of microbial cultures was checked by assessing the cultural characteristics (turbidity, deposits, surface formations) at 20, 44, and 72 h.

## 4. Conclusions

The present study aimed to develop Fe_3_O_4_@SiO_2_ core–shell nanoparticles as efficient carriers for the delivery and controlled release of three types of EOs, namely thyme, rosemary, and basil. The Fe_3_O_4_@SiO_2_ core–shell nanoparticles were synthesized through the co-precipitation and microwave-assisted hydrothermal methods in order to establish the influence of the synthesis method on the properties of the nanosystems. In this context, it was concluded that the hydrothermal method leads to a higher percentage of silica within the nanosystems, which could be attributed to the higher surface reactivity of the magnetite cores. Additionally, due to an increased crystal growth associated with this method, the size of the magnetite nanoparticles was larger, as confirmed by XRD, TEM, SEM, and DLS. The core–shell nanoparticles maintained their magnetic properties at a level proportional to the mass fraction of the magnetite cores. The attachment of the EOs was successful in all cases, with a higher yield for thyme EO, which also explains the antimicrobial activity for these samples. While the hydrothermal method allowed for a higher loading efficiency of the EOs, it resulted in reduced release, as shown by the antimicrobial studies. Thus, the efficiency of the Fe_3_O_4_@SiO_2_@thyme nanosystems was demonstrated. Thus, they could successfully be applied in a variety of antimicrobial applications, either biomedical e.g., coatings, wound dressings, or for food packaging.

## Figures and Tables

**Figure 1 antibiotics-10-01138-f001:**
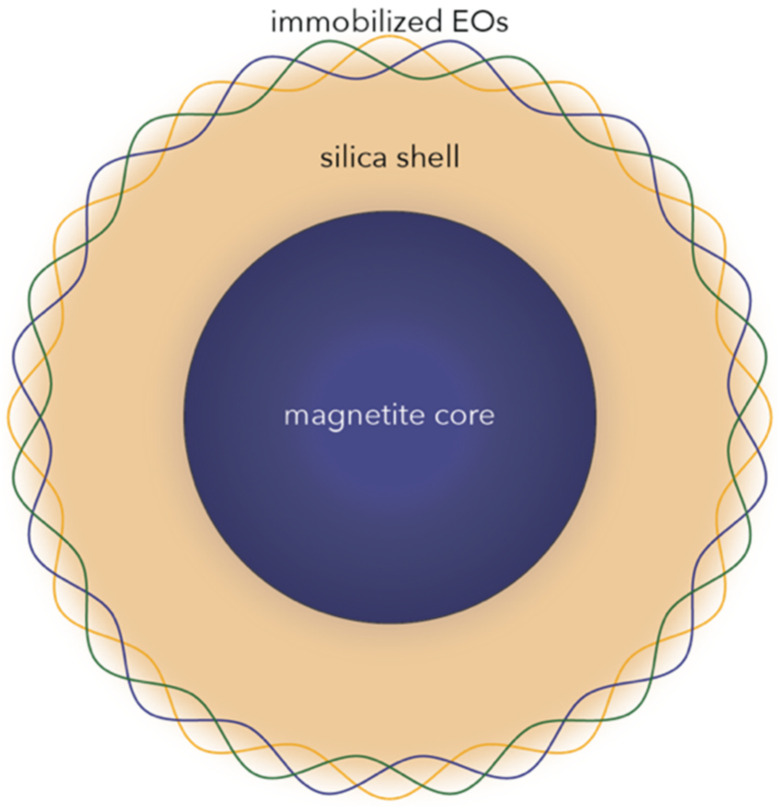
Schematic representation of the core–shell nanosystems investigated in this study.

**Figure 2 antibiotics-10-01138-f002:**
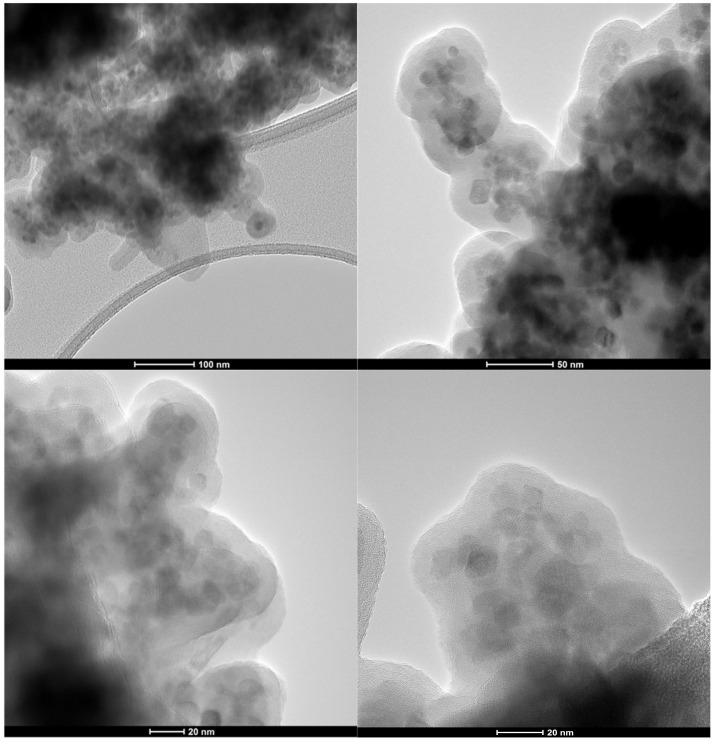
TEM images of the synthesized core–shell Fe_3_O_4_@SiO_2_ nanostructures at different scale bars.

**Figure 3 antibiotics-10-01138-f003:**
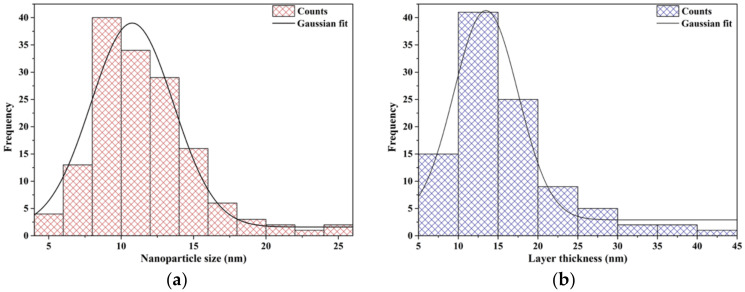
The size distributions associated with the size of the magnetite cores (**a**) and thickness of the silica layer (**b**).

**Figure 4 antibiotics-10-01138-f004:**
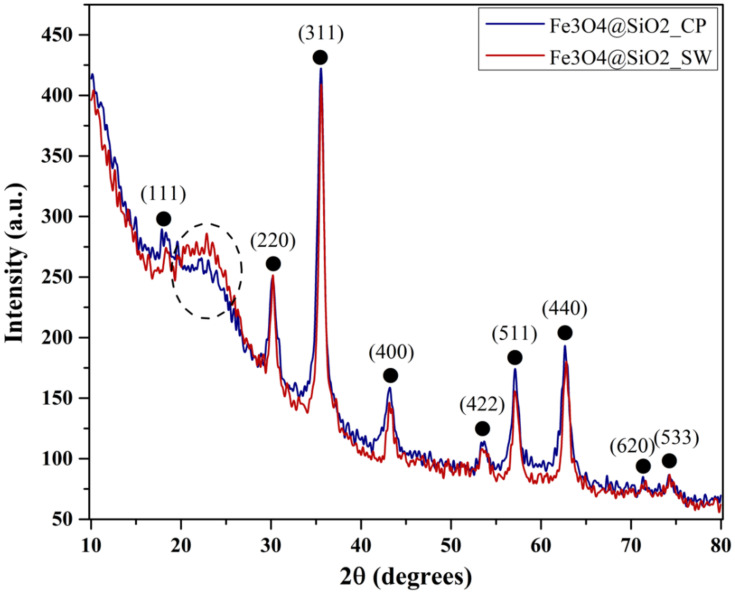
Diffractograms for the Fe_3_O_4_@SiO_2__CP and Fe_3_O_4_@SiO_2__SW samples (●—Fe_3_O_4_).

**Figure 5 antibiotics-10-01138-f005:**
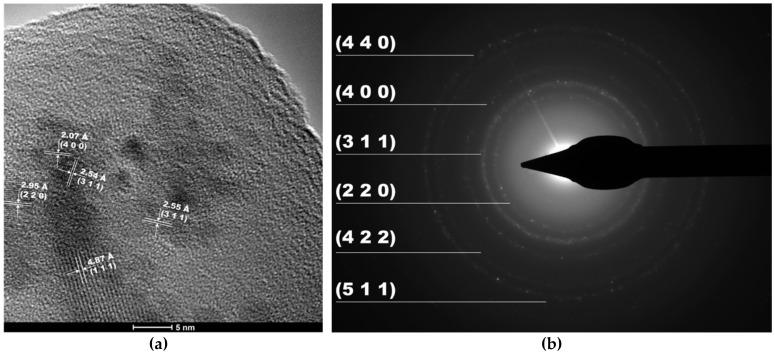
HR-TEM image (**a**), SAED diffraction pattern (**b**), and the corresponding Miller indices for the Fe_3_O_4_@SiO_2__CP core–shell nanosystems.

**Figure 6 antibiotics-10-01138-f006:**
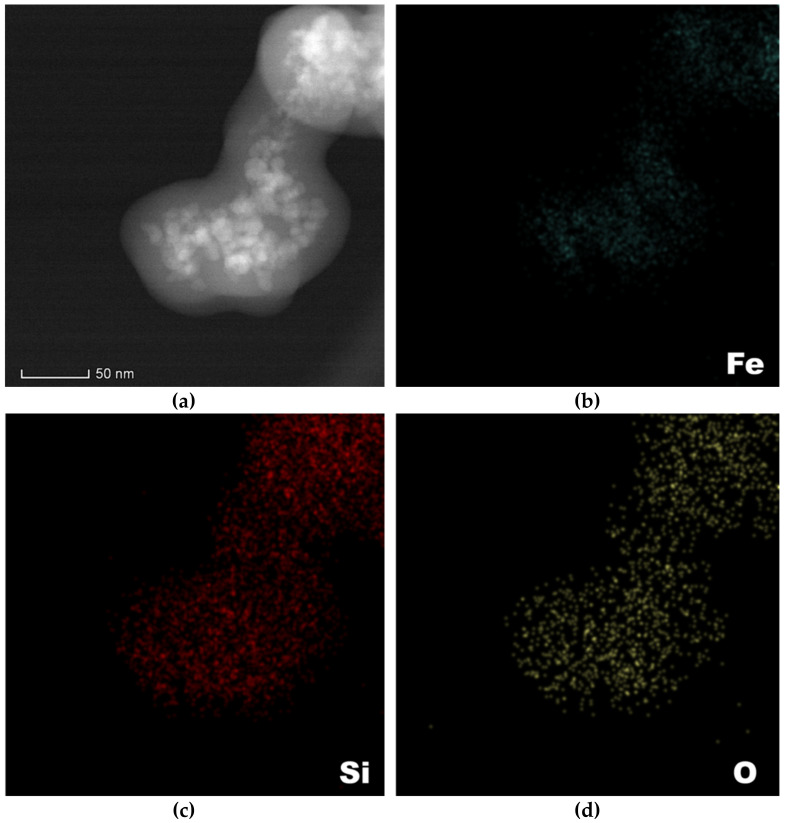
Elemental mapping of the Fe_3_O_4_@SiO_2__CP core–shell nanosystems: (**a**)—mapped area, (**b**)—Fe, (**c**)—Si, and (**d**)—O.

**Figure 7 antibiotics-10-01138-f007:**
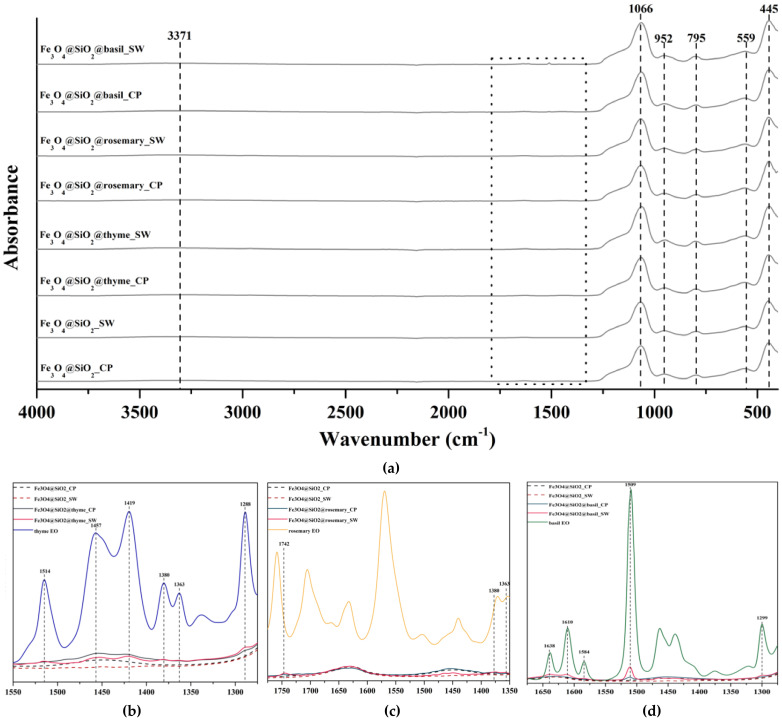
FT-IR spectra for (**a**) all eight samples, (**b**) thyme EO-functionalized samples and thyme EO; (**c**) rosemary EO-functionalized samples and rosemary EO; (**d**) basil EO-functionalized samples and basil EO.

**Figure 8 antibiotics-10-01138-f008:**
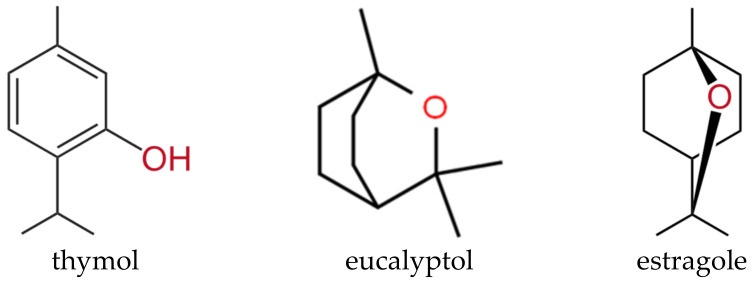
The chemical structures of the major compounds identified in the EO-functionalized nanosystems, namely thymol, eucalyptol, and estragole.

**Figure 9 antibiotics-10-01138-f009:**
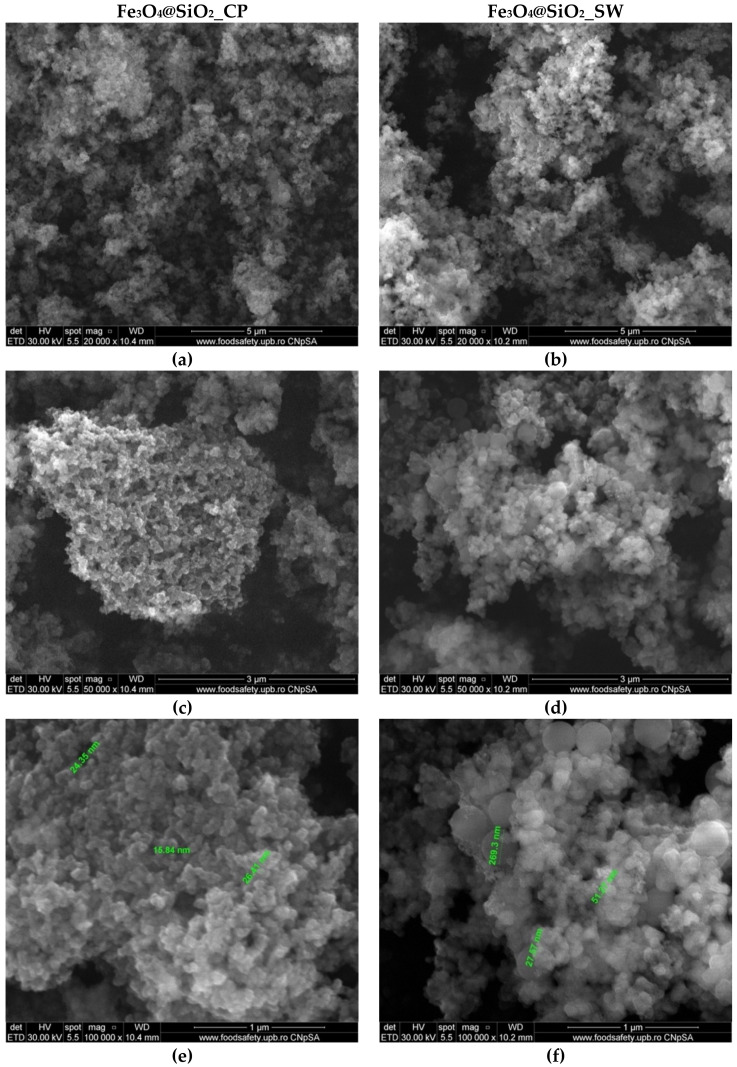
SEM images at three different magnifications ((**a**,**b**)—20,000×, (**c**,**d**)—50,000×, and (**e**,**f**)—100,000×) for the Fe_3_O_4_@SiO_2__CP samples (**left**) and Fe_3_O_4_@SiO_2__SW samples (**right**).

**Figure 10 antibiotics-10-01138-f010:**
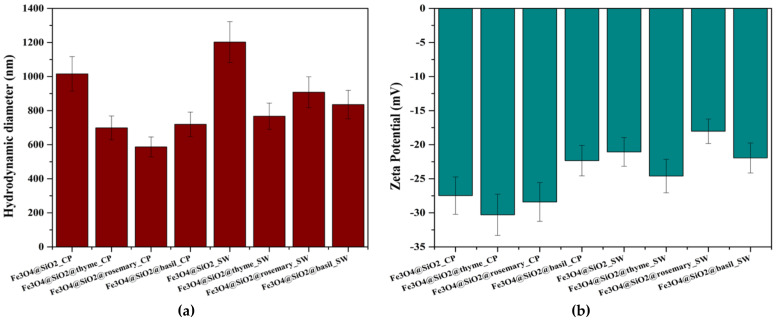
Representation of the mean hydrodynamic diameter (**a**) and zeta potential (**b**) values as a comparison between all eight samples.

**Figure 11 antibiotics-10-01138-f011:**
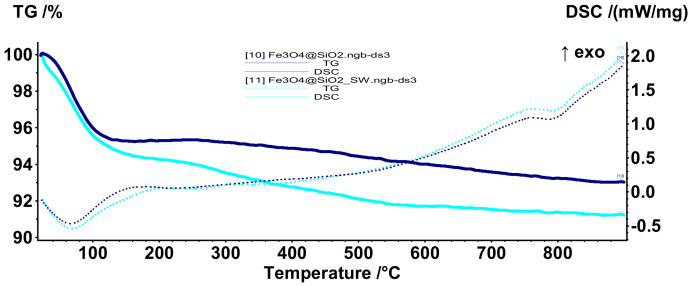
TG-DSC curves for the Fe_3_O_4_@SiO_2__CP and Fe_3_O_4_@SiO_2__SW nanosystems.

**Figure 12 antibiotics-10-01138-f012:**
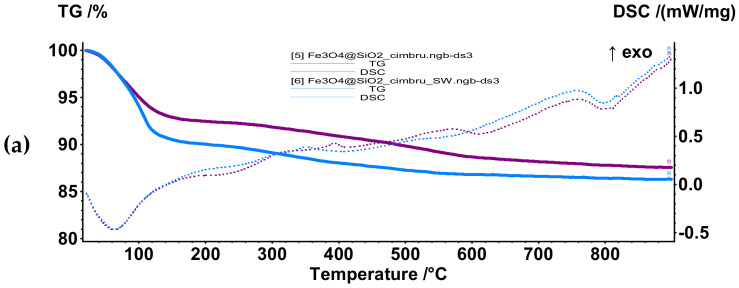
TG-DSC curves for samples (**a**) Fe_3_O_4_@SiO_2_@thyme_CP and Fe_3_O_4_@SiO_2_@thyme_SW; (**b**) Fe_3_O_4_@SiO_2_@rosemary_CP and Fe_3_O_4_@SiO_2_@rosemary_SW, (**c**) Fe_3_O_4_@SiO_2_@basil_CP and Fe_3_O_4_@SiO_2_@basil_SW.

**Figure 13 antibiotics-10-01138-f013:**
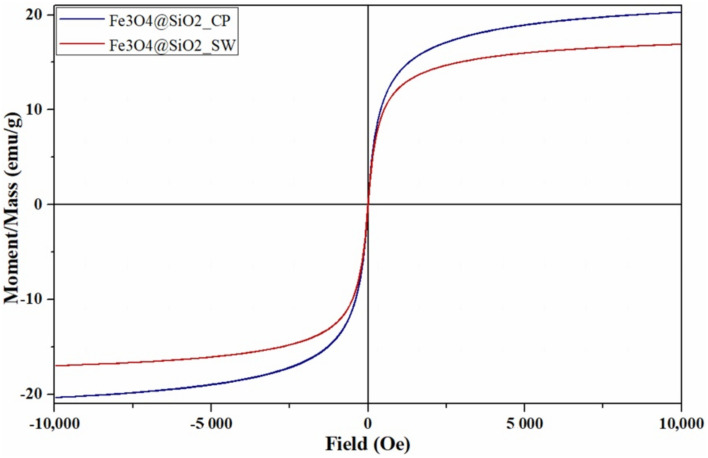
Field-dependent magnetization measurements for Fe_3_O_4_@SiO_2_ nanosystems synthesized through co-precipitation and microwave-assisted hydrothermal method.

**Table 1 antibiotics-10-01138-t001:** Summary of the nanosystems obtained depending on the synthesis method and the type of EOs immobilized.

Synthesis Method	Sample	Code
Co-precipitation method	Fe_3_O_4_@SiO_2_	Fe_3_O_4_@SiO_2__CP
	Fe_3_O_4_@SiO_2_@thyme EO	Fe_3_O_4_@SiO_2_@thyme_CP
	Fe_3_O_4_@SiO_2_@rosemary EO	Fe_3_O_4_@SiO_2_@rosemary_CP
	Fe_3_O_4_@SiO_2_@basil EO	Fe_3_O_4_@SiO_2_@basil_CP
Microwave-assisted hydrothermal method	Fe_3_O_4_@SiO_2_	Fe_3_O_4_@SiO_2__SW
	Fe_3_O_4_@SiO_2_@thyme EO	Fe_3_O_4_@SiO_2_@thyme_SW
	Fe_3_O_4_@SiO_2_@rosemary EO	Fe_3_O_4_@SiO_2_@rosemary_SW
	Fe_3_O_4_@SiO_2_@basil EO	Fe_3_O_4_@SiO_2_@basil_SW

**Table 2 antibiotics-10-01138-t002:** The crystallite size values for the Fe_3_O_4_@SiO_2__CP and Fe_3_O_4_@SiO_2__SW samples, calculated using the information provided by the diffractograms.

Sample	hkl	K	λ (Å)	FWHM (rad)	2θ (°)	D (nm)	D Average (nm)
**Fe_3_O_4_@SiO_2__CP**	111	0.9	1.5418	0.0180	18.3894	7.80	9.31
	220			0.0160	30.1755	8.94	
	311			0.0151	35.5235	9.59	
	400			0.0192	43.1278	7.75	
	422			0.0157	53.6131	9.86	
	511			0.0142	57.1491	11.11	
	440			0.0188	62.6989	8.60	
**Fe_3_O_4_@SiO_2__SW**	111			0.0124	18.4202	11.31	11.16
	220			0.0097	30.1963	14.72	
	311			0.0112	35.6044	12.98	
	400			0.0141	43.2380	10.51	
	422			0.0215	53.6705	7.20	
	511			0.0152	57.2089	10.37	
	440			0.0147	62.7988	11.02	

**Table 3 antibiotics-10-01138-t003:** Compounds identified from the thyme, rosemary, and basil EOs and their retention time from the GC-MS chromatograms.

Sample	Compound	Retention Time
Fe_3_O_4_@SiO_2_@thyme_CP/Fe_3_O_4_@SiO_2_@thyme_SW	p-cymene	12.785
	eucalyptol	13.056
	linalyl formate	15.557
	endo-borneol	18.162
	terpinen-4-ol	18.475
	α-terpineol	19.006
	p-cymen-7-ol	22.361
	thymol	22.466
	phenanthrenone	46.422
Fe_3_O_4_@SiO_2_@rosemary_CP/Fe_3_O_4_@SiO_2_@rosemary_SW	eucalyptol	13.059
	(+)-2-bornanone	17.300
	endo-borneol	18.206
	α-terpineol	19.129
	β-longipinene	26.746
	thymol	22.439
Fe_3_O_4_@SiO_2_@basil_CP/Fe_3_O_4_@SiO_2_@basil_SW	trans-linalool oxide (furanoid)	15.069
	trans-linalyl formate	15.587
	linalool oxide (pyranoid)	18.112
	levomenthol	18.403
	estragole	19.143

**Table 4 antibiotics-10-01138-t004:** Bond characteristics for the absorption bands registered in FT-IR spectra.

Type of Bond	Wavenumber (cm^−1^)
C-O stretching	1288, 1299
O-H bending (phenol)	1363, 1380
O-H bending (alcohol)	1419
C-H bending	1457
C=C stretching	1509, 1514, 1610, 1638
aromatic C-C stretching	1584
C=O stretching	1742

**Table 5 antibiotics-10-01138-t005:** The hydrodynamic diameter and zeta potential values measured for the core–shell iron oxide–silica nanoparticles.

Scheme	Hydrodynamic Diameter (nm)	Zeta Potential (mV)
Fe_3_O_4_@SiO_2__CP	1015.36	−27.47
Fe_3_O_4_@SiO_2_@thyme_CP	698.76	−30.29
Fe_3_O_4_@SiO_2_@rosemary_CP	586.98	−28.41
Fe_3_O_4_@SiO_2_@basil_CP	719.30	−22.34
Fe_3_O_4_@SiO_2__SW	1202.38	−21.06
Fe_3_O_4_@SiO_2_@thyme_SW	767.24	−24.60
Fe_3_O_4_@SiO_2_@rosemary_SW	908.08	−18.03
Fe_3_O_4_@SiO_2_@basil_SW	835.68	−21.95

**Table 6 antibiotics-10-01138-t006:** The mass losses registered and the associated thermal effects according to the TG-DSC curves.

Sample	Mass Loss (%)				Thermal Effects (°C)		Loading Efficiency (%)
	20–180 °C	180–500 °C	500–900 °C	Residual Mass	Endothermic	Exothermic	
Fe_3_O_4_@SiO_2__CP	4.69	0.84	1.41	93.03	66.4	-	-
Fe_3_O_4_@SiO_2_@thyme_CP	7.45	2.72	2.29	87.55	62.4	393.2	2.76
Fe_3_O_4_@SiO_2_@rosemary_CP	6.30	2.19	1.71	89.80	61.7	387.6	1.61
Fe_3_O_4_@SiO_2_@basil_CP	6.67	2.33	1.73	89.24	61.8	390.3	1.98
Fe_3_O_4_@SiO_2__SW	5.66	2.24	0.87	91.23	68.5	-	-
Fe_3_O_4_@SiO_2_@thyme_SW	9.84	2.88	0.97	86.30	64.2	350.9	4.18
Fe_3_O_4_@SiO_2_@rosemary_SW	7.96	2.82	0.97	88.48	67.2	348.9	2.30
Fe_3_O_4_@SiO_2_@basil_SW	8.96	2.94	0.99	87.09	61.8	347.0	3.30

**Table 7 antibiotics-10-01138-t007:** The magnetic properties of the Fe_3_O_4_@SiO_2_ nanosystems synthesized through co-precipitation and the microwave-assisted hydrothermal method.

Sample	M_s_ (emu/g)	M_r_ (emu/g)	H_c_ (Oe)
Fe_3_O_4_@SiO_2__CP	20.323	0.193	4.158
Fe_3_O_4_@SiO_2__SW	16.954	0.386	8.942

**Table 8 antibiotics-10-01138-t008:** The antimicrobial activity of the core–shell nanoparticles at three different concentrations (1, 2, and 4 mg/mL) and of the thyme, rosemary, and basil EOs against *Staphylococcus aureus*, *Pseudomonas* aeruginosa, *Escherichia coli*, and *Candida albicans* after 72 h incubation.

Sample	Concentration	Microbial Species				Control	
		*Staphylococcus aureus* ATCC 25923 (2 × 10^6^ UFC/mL Test)	*Pseudomonas aeruginosa* ATCC 27853 (4 × 10^5^ UFC/mL Test)	*Escherichia coli* ATCC 25922 (3.6 × 10^6^ UFC/mL Test)	*Candida albicans* ATCC 10231 (1 × 10^5^ UFC/mL Test)	BHI	Sabouraud
Fe_3_O_4_@SiO_2__CP	1 mg/mL	+	+	+	+	-	-
	2 mg/mL	ND	ND	ND	ND	-	-
	4 mg/mL	+	+	+	+	-	-
Fe_3_O_4_@SiO_2_@thyme_CP	1 mg/mL	+	+	+	+	-	-
	2 mg/mL	-	ND	-	-	-	-
	4 mg/mL	-	+	-	-	-	-
Fe_3_O_4_@SiO_2_@rosemary_CP	1 mg/mL	+	+	+	+	-	-
	2 mg/mL	ND	ND	ND	ND	-	-
	4 mg/mL	+	+	+	+	-	-
Fe_3_O_4_@SiO_2_@basil_CP	1 mg/mL	+	+	+	+	-	-
	2 mg/mL	ND	ND	ND	ND	-	-
	4 mg/mL	+	+	+	+	-	-
Fe_3_O_4_@SiO_2__SW	1 mg/mL	+	+	+	+	-	-
	2 mg/mL	ND	ND	ND	ND	-	-
	4 mg/mL	+	+	+	+	-	-
Fe_3_O_4_@SiO_2_@thyme_SW	1 mg/mL	+	+	+	+	-	-
	2 mg/mL	-	ND	+	+	-	-
	4 mg/mL	-	+	-	-	-	-
Fe_3_O_4_@SiO_2_@rosemary_SW	1 mg/mL	+	+	+	+	-	-
	2 mg/mL	ND	ND	ND	ND	-	-
	4 mg/mL	+	+	+	+	-	-
Fe_3_O_4_@SiO_2_@basil_SW	1 mg/mL	+	+	+	+	-	-
	2 mg/mL	ND	ND	ND	ND	-	-
	4 mg/mL	+	+	+	+	-	-
thyme EO	0.1 µL/mL	+	+	+	+	-	-
	0.2 µL/mL	+	ND	+	+	-	-
	0.4 µL/mL	-	+	+	+	-	-
rosemary EO	0.1 µL/mL	+	+	+	+	-	-
	0.2 µL/mL	ND	ND	ND	ND	-	-
	0.4 µL/mL	+	+	+	+	-	-
basil EO	0.1 µL/mL	+	+	+	+	-	-
	0.2 µL/mL	ND	ND	ND	ND	-	-
	0.4 µL/mL	+	+	+	+	-	-
BHI	-	+	+	+	ND	-	ND
Sabouraud	-	ND	ND	ND	+	ND	-

+ refers to presence of turbidity and/or deposits, microbial culture present; -: refers to absence of turbidity and/or deposits, microbial culture absent; ND: not determined; BHI: Brain Heart Infusion medium.

**Table 9 antibiotics-10-01138-t009:** MIC values of thymol, eucalyptol, and estragole, the major compounds found within thyme, rosemary, basil EOs, respectively, from the available literature.

Compound	Microbial Species				Ref.
	*Staphylococcus aureus*	*Pseudomonas aeruginosa*	*Escherichia coli*	*Candida albicans*	
thymol	MIC: 0.2 mg/mL	MIC: 0.4 mg/mL	MIC: 0.2 mg/mL	MIC: 0.03 mg/mL	[76,79,80]
eucalyptol	MIC: 0.5 mg/mL	MIC: -	MIC: 0.5 mg/mL	MIC: 0.03 mg/mL	[74]
estragole	MIC: 1.37 mg/mL	MIC: -	MIC: 2.7 mg/mL	MIC: 0.75 mg/mL	[81]

## Data Availability

Data is contained within the article or Appendix A.

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
