# Peer review of "Iron Oxide–Silica Core–Shell Nanoparticles Functionalized with Essential Oils for Antimicrobial Therapies"

_antibiotics, 2021, doi:10.3390/antibiotics10091138_

Round 1
Reviewer 1 Report
I have evaluated the manuscript (Antibiotics-1355216) titled “Iron Oxide-Silica Core-Shell Nanoparticles Functionalized with Essential Oils for Antimicrobial Therapies” by Chircov and coworkers, describing the antimicrobial activity of essential oils immobilized with iron oxide-silica nanoparticles.
I am glad to review this manuscript, covering important aspects of the subject with excellent presentation of results and clearly describing the outcome. All standard methods were used for the experiments. I found the document interesting for the readers and follow the scope of the journal Antibiotics.
I would recommend the article could be published in Antibiotics, after a minor revision. There are technical errors, I hope the editor will take care of them.
The authors need to address the below-mentioned queries.
- Line 24: “this paper” to this manuscript”.
- Line 25: “Fe3O4@SiO2” to “Fe3O4@SiO2” and correct other similar errors.
- In the introduction, the author could provide the background of nanotechnology-based approaches as antimicrobial.
- The percentage of major chemical constituents of thyme, rosemary, and basil EOs in the table could be useful to get a better understanding of active chromophore, if possible main functional groups of each class could be presented.
- Any advantage of methods to make Fe3O4@SiO2 nanosystems used by author over other available methods.
- The author needs to discuss the effect of the size and thickness of nanoparticles on antimicrobial activity.
- For the antimicrobial activity of Iron Oxide-Silica Core-Shell Nanoparticles Functionalized with Essential Oils, control is missing.
- Space is missing for all temperatures, e.g “80°C”.
- The author could provide a figure showing nanoparticles core, surface, and immobilize oil.The author could include the following references:
- Arias, L.S.; Pessan, J.P.; Vieira, A.P.M.; Lima, T.M.T.d.; Delbem, A.C.B.; Monteiro, D.R. Iron Oxide Nanoparticles for Biomedical Applications: A Perspective on Synthesis, Drugs, Antimicrobial Activity, and Toxicity. Antibiotics 2018, 7, 46. https://doi.org/10.3390/antibiotics7020046.
- Amedea B. Seabra, Milena T. Pelegrino, Paula S. Haddad, Antimicrobial Applications of Superparamagnetic Iron Oxide Nanoparticles: Perspectives and Challenges, Nanostructures for Antimicrobial Therapy, 2017, 531-550. https://doi.org/10.1016/B978-0-323-46152-8.00024-X.
- Makabenta, J.M.V., Nabawy, A., Li, CH. et al. Nanomaterial-based therapeutics for antibiotic-resistant bacterial infections. Nat Rev Microbiol 19, 23–36 (2021). https://doi.org/10.1038/s41579-020-0420-1
Author Response
Reviewer 1
I have evaluated the manuscript (Antibiotics-1355216) titled “Iron Oxide-Silica Core-Shell Nanoparticles Functionalized with Essential Oils for Antimicrobial Therapies” by Chircov and coworkers, describing the antimicrobial activity of essential oils immobilized with iron oxide-silica nanoparticles.
I am glad to review this manuscript, covering important aspects of the subject with excellent presentation of results and clearly describing the outcome. All standard methods were used for the experiments. I found the document interesting for the readers and follow the scope of the journal Antibiotics.
Thank you very much for your review. We hope that we met all your suggestions and comments through the modifications that were made to the manuscript.
I would recommend the article could be published in Antibiotics, after a minor revision. There are technical errors, I hope the editor will take care of them.
The authors need to address the below-mentioned queries.
- Line 24: “this paper” to this manuscript”.
The modification was made.
- Line 25: “Fe3O4@SiO2” to “Fe3O4@SiO2” and correct other similar errors.
The issue was fixed, and the entire manuscript was checked for such errors.
- In the introduction, the author could provide the background of nanotechnology-based approaches as antimicrobial.
The introduction was improved accordingly.
- The percentage of major chemical constituents of thyme, rosemary, and basil EOs in the table could be useful to get a better understanding of active chromophore, if possible main functional groups of each class could be presented.
The chemical structure for each compound was added and the functional group was highlighted.
- Any advantage of methods to make Fe3O4@SiO2 nanosystems used by author over other available methods.
The advantages and disadvantages for each method were described.
- The author needs to discuss the effect of the size and thickness of nanoparticles on antimicrobial activity.
The effects of size and thickness of the core-shell nanosystems on the antimicrobial activity were discussed.
- For the antimicrobial activity of Iron Oxide-Silica Core-Shell Nanoparticles Functionalized with Essential Oils, control is missing.
The control was added to the Table.
- Space is missing for all temperatures, e.g “80°C”.
The issue was fixed.
- The author could provide a figure showing nanoparticles core, surface, and immobilize oil.The author could include the following references:
- Arias, L.S.; Pessan, J.P.; Vieira, A.P.M.; Lima, T.M.T.d.; Delbem, A.C.B.; Monteiro, D.R. Iron Oxide Nanoparticles for Biomedical Applications: A Perspective on Synthesis, Drugs, Antimicrobial Activity, and Toxicity. Antibiotics 2018, 7, 46. https://doi.org/10.3390/antibiotics7020046.
- Amedea B. Seabra, Milena T. Pelegrino, Paula S. Haddad, Antimicrobial Applications of Superparamagnetic Iron Oxide Nanoparticles: Perspectives and Challenges, Nanostructures for Antimicrobial Therapy, 2017, 531-550. https://doi.org/10.1016/B978-0-323-46152-8.00024-X.
- Makabenta, J.M.V., Nabawy, A., Li, CH. et al. Nanomaterial-based therapeutics for antibiotic-resistant bacterial infections. Nat Rev Microbiol 19, 23–36 (2021). https://doi.org/10.1038/s41579-020-0420-1
A figure showing the structure of the core-shell systems and the references were added to the manuscript.
Reviewer 2 Report
The authors describe the synthesis, loading, and antimicrobial properties of core-shell magnetic iron-silica nanoparticles loaded with compounds from essential oils. The particles are extensively characterised to a high level of detail and I think the paper has great potential. However, the antimicrobial properties are somewhat ambiguous and require further investigation and discussion. I think the manuscript would be greatly improved by including the following:
- The authors should describe which components of the essential oil (EO) mixtures (shown in Table 1) are actively antimicrobial. They should then assign the FT-IR spectra to functional groups found within each of the components in each EO mixture. Particularly for rosemary, it's clear from the TGA that some loading has occurred but key peaks are missing from the spectrum (e.g. large peak at 1575 cm-1 - what does this correspond to? Whatever this compound is has not been loaded). If the authors can rule out whether the missing peaks come from active antimicrobial components of the EOs, or if they are inert compounds, this would enable better discussion of the antimicrobial properties.
- The only system that really shows any antimicrobial properties is thyme-loaded nanoparticles. What is the MIC of pure thyme alone? The authors should investigate a greater range of concentrations to determine the true molar MIC. This can then be discussed in terms of the loaded molar amount of thyme in the nanoparticles. Does loading the same molar amount of thyme improve its properties? Since thyme is a mixture, perhaps we need to consider what component would be the most antimicrobial, corrected by its relative concentration within the thyme mixture. This could be compared to both the loading% and the relative loaded composition from the FT-IR spectrum to obtain a molar concentration of this component within the nanoparticle solution, which could be compared to free drug.
- It's strange that the co-precipitation method produced nanoparticles that had lower loading but greater antimicrobial properties than those produced by microwave synthesis. The authors conclude (line 476) that this is because of different release rates but don't show any release data. It could instead be that the different synthesis routes led to different relative distributions of loaded compounds from the EOs. If co-precipitation shows relatively more antimicrobial compounds loaded (from the IR spectrum and known functional groups of active/inactive compounds), this would explain this contradictory result.
- In all cases (rosemary, thyme, and basil) the authors look at the antimicrobial behaviour of three mass concentrations, however it's not clear how these relate to the mass concentrations of the loaded compounds in the nanoparticle solutions. The table makes it look like some nanoparticles outperform the EOs alone, but I don't think we can know this without giving context into how much EO was present in the nanoparticle solutions. In the table, the authors should quote the mass concentration of nanoparticles, and also the mass concentration of loaded EO (= total mass x mass% of loaded EO), perhaps as a separate column.
- In the experimental section (line 388), it seems that there was no purification of the nanoparticles after loading with the EOs. Is this true or does this section need to be rewritten? If no purification occurred, the characterisation data does not make sense as the loaded and unloaded drug would both be seen in the spectrum, however I assume some purification took place.
- As a general comment, the antimicrobial R+D section (line 303 onwards) is confusing and much better described by the table itself. In my opinion, this discussion talking through the table doesn't add much because there is no discussion of why these results might have occurred. I think a lot of this discussion could be stripped back to make room for the discussion resulting from the outcomes of the above points 1-4, which I believe would greatly improve this section.
More minor comments:
- The sample labelling convention is outlined in Table 8, but this should be either placed earlier, or briefly described at the start of the results and discussion. What does "SW" stand for if it refers to "microwave-assisted hydrothermal method"?
- Figure 2 - are these solid lines fits? What is the fitting function? On line 118, the authors say "most" of the particles fall within a certain size range. Perhaps it would be more scientific to say the range at FWHM of the solid line (assuming it was a fit and not just a line to guide the eye).
- Figure 7 - "magnetization" should be "magnification"
- Figure 8 - the size distribution and correlation functions of the DLS data should be placed in the SI.
Author Response
Reviewer 2
The authors describe the synthesis, loading, and antimicrobial properties of core-shell magnetic iron-silica nanoparticles loaded with compounds from essential oils. The particles are extensively characterised to a high level of detail and I think the paper has great potential. However, the antimicrobial properties are somewhat ambiguous and require further investigation and discussion. I think the manuscript would be greatly improved by including the following:
Thank you very much for your review. We hope that the quality of the manuscript was improved according to your comments and suggestions.
- The authors should describe which components of the essential oil (EO) mixtures (shown in Table 1) are actively antimicrobial. They should then assign the FT-IR spectra to functional groups found within each of the components in each EO mixture. Particularly for rosemary, it's clear from the TGA that some loading has occurred but key peaks are missing from the spectrum (e.g. large peak at 1575 cm-1 - what does this correspond to? Whatever this compound is has not been loaded). If the authors can rule out whether the missing peaks come from active antimicrobial components of the EOs, or if they are inert compounds, this would enable better discussion of the antimicrobial properties.
The chemical structure for each compound was added and the functional group was highlighted. Additionally, the discussion on the results from the FT-IR spectra was improved in order to refer to the compounds of each essential oil used.
- The only system that really shows any antimicrobial properties is thyme-loaded nanoparticles. What is the MIC of pure thyme alone? The authors should investigate a greater range of concentrations to determine the true molar MIC. This can then be discussed in terms of the loaded molar amount of thyme in the nanoparticles. Does loading the same molar amount of thyme improve its properties? Since thyme is a mixture, perhaps we need to consider what component would be the most antimicrobial, corrected by its relative concentration within the thyme mixture. This could be compared to both the loading% and the relative loaded composition from the FT-IR spectrum to obtain a molar concentration of this component within the nanoparticle solution, which could be compared to free drug.
The MIC value of the thyme EO was determined at 0.4 µL/mL, since the concentration of 0.2 µL/mL did not inhibit the growth of the microbial species. Thus, we have the demonstrated the synergistic antimicrobial effects of the core-shell systems functionalized with thyme EO.
- It's strange that the co-precipitation method produced nanoparticles that had lower loading but greater antimicrobial properties than those produced by microwave synthesis. The authors conclude (line 476) that this is because of different release rates but don't show any release data. It could instead be that the different synthesis routes led to different relative distributions of loaded compounds from the EOs. If co-precipitation shows relatively more antimicrobial compounds loaded (from the IR spectrum and known functional groups of active/inactive compounds), this would explain this contradictory result.
These results were better explained according to your suggestions.
- In all cases (rosemary, thyme, and basil) the authors look at the antimicrobial behaviour of three mass concentrations, however it's not clear how these relate to the mass concentrations of the loaded compounds in the nanoparticle solutions. The table makes it look like some nanoparticles outperform the EOs alone, but I don't think we can know this without giving context into how much EO was present in the nanoparticle solutions. In the table, the authors should quote the mass concentration of nanoparticles, and also the mass concentration of loaded EO (= total mass x mass% of loaded EO), perhaps as a separate column.
The concentrations for the EOs were selected as the mass concentration from each sample. Specifically, if 100 µL were added to 1 g of the nanoparticles, it means that in 1 mg of nanoparticles there would be 0.1 µL of EO and so on for 2 mg and 4 mg.
- In the experimental section (line 388), it seems that there was no purification of the nanoparticles after loading with the EOs. Is this true or does this section need to be rewritten? If no purification occurred, the characterisation data does not make sense as the loaded and unloaded drug would both be seen in the spectrum, however I assume some purification took place.
Indeed, there was no purification of the nanosystems after EO loading. The free bioactive molecules do not appear in the spectra due to the high volatilization which is characteristic for EOs.
- As a general comment, the antimicrobial R+D section (line 303 onwards) is confusing and much better described by the table itself. In my opinion, this discussion talking through the table doesn't add much because there is no discussion of why these results might have occurred. I think a lot of this discussion could be stripped back to make room for the discussion resulting from the outcomes of the above points 1-4, which I believe would greatly improve this section.
The section regarding the antimicrobial results was summarized and the results were better discussed according to your suggestions.
More minor comments:
- The sample labelling convention is outlined in Table 8, but this should be either placed earlier, or briefly described at the start of the results and discussion. What does "SW" stand for if it refers to "microwave-assisted hydrothermal method"?
SW stands for Synthwave, which is the equipment used for the microwave-assisted hydrothermal method. The labeling was better explained in the manuscript.
- Figure 2 - are these solid lines fits? What is the fitting function? On line 118, the authors say "most" of the particles fall within a certain size range. Perhaps it would be more scientific to say the range at FWHM of the solid line (assuming it was a fit and not just a line to guide the eye).
The fitting function used was added and the average nanoparticle size and shell thickness was modified according to your suggestions.
- Figure 7 - "magnetization" should be "magnification"
The issue was fixed.
- Figure 8 - the size distribution and correlation functions of the DLS data should be placed in the SI.
The size distribution for each sample and the associated correlation graph were added to the Supplementary Material.
Reviewer 3 Report
The present manuscript entitled “Iron Oxide-Silica Core-Shell Nanoparticles Functionalized with Essential Oils for Antimicrobial Therapies” authored by Cristina Chircov et al. describes the developing nanostructured systems based on Fe3O4@SiO2 core-shell nanoparticles and three different types of essential oils, i.e., thyme, rosemary, and basil. Furthermore, the authors have also demonstrated the comparative study between co-precipitation and microwave-assisted hydro-thermal methods for the synthesis of Fe3O4@SiO2 core-shell nanoparticles. As-synthesized samples are investigated various characterization techniques such as XRD, FT-IR, DLS, TG-DSC, SEM, and TEM etc., Additionally, the as-synthesized nanocomposites were assessed through in vitro tests on S. aureus, P. aeruginosa, E. coli, and C. albicans. It is a well-organized article and lack of major errors. Therefore, I recommend it for publication. However, certain Minor issues are detailed below which need to be addressed before its final acceptance in the Antibiotics.
I advise the authors to take the following points into account while revising their manuscript.
Comment 1: Firstly, I would like to draw the attention of the authors that I found there are some typographical errors in the manuscript, so authors need to correct them in the revised manuscript. For e.g. In line 25, “Fe3O4@SiO2” should be “Fe3O4@SiO2”; Line 350, “FeSO4·7H2O” should be “FeSO4·7H2O”; line 351 “FeCl3·6H2O” should be “FeCl3·6H2O”; line 351 “NH4OH” should be “NH4OH” etc.,
Comment 2: Change the Figure 1 caption, as “TEM images of the synthesized core-shell Fe3O4@SiO2 nanostructures at different scale bars.
Comment 3: The TEM and SEM results explanation should explore and discuss better their results with some more references in order to prepare a better discussion.
Author Response
The present manuscript entitled “Iron Oxide-Silica Core-Shell Nanoparticles Functionalized with Essential Oils for Antimicrobial Therapies” authored by Cristina Chircov et al. describes the developing nanostructured systems based on Fe3O4@SiO2 core-shell nanoparticles and three different types of essential oils, i.e., thyme, rosemary, and basil. Furthermore, the authors have also demonstrated the comparative study between co-precipitation and microwave-assisted hydro-thermal methods for the synthesis of Fe3O4@SiO2 core-shell nanoparticles. As-synthesized samples are investigated various characterization techniques such as XRD, FT-IR, DLS, TG-DSC, SEM, and TEM etc., Additionally, the as-synthesized nanocomposites were assessed through in vitro tests on S. aureus, P. aeruginosa, E. coli, and C. albicans. It is a well-organized article and lack of major errors. Therefore, I recommend it for publication. However, certain Minor issues are detailed below which need to be addressed before its final acceptance in the Antibiotics.
Thank you very much for your review. We modified the manuscript according to your comments and suggestions.
I advise the authors to take the following points into account while revising their manuscript.
Comment 1: Firstly, I would like to draw the attention of the authors that I found there are some typographical errors in the manuscript, so authors need to correct them in the revised manuscript. For e.g. In line 25, “Fe3O4@SiO2” should be “Fe3O4@SiO2”; Line 350, “FeSO4·7H2O” should be “FeSO4·7H2O”; line 351 “FeCl3·6H2O” should be “FeCl3·6H2O”; line 351 “NH4OH” should be “NH4OH” etc.,
The errors were fixed throughout the manuscript.
Comment 2: Change the Figure 1 caption, as “TEM images of the synthesized core-shell Fe3O4@SiO2 nanostructures at different scale bars.
The caption was modified accordingly.
Comment 3: The TEM and SEM results explanation should explore and discuss better their results with some more references in order to prepare a better discussion.
The discussion regarding TEM and SEM results was improved according to your suggestion.
Round 2
Reviewer 2 Report
After the rapid additions from the authors, I am unfortunately left feeling that there are more issues with this article than before.
- Having the structures in the table is helpful. However, my comments surrounding the MIC of the antimicrobial compounds that make up each of the EO mixtures has not been adequately addressed. For example, what was the MIC of thymol, p-cymene, linalool, borneol, etc. against each of the pathogens? How does this compare to the MIC of the particles (containing a loaded mixture of these compounds) and to the MIC of the individually loaded compounds on the particles (calculated from FTIR and TGA)?
- There was no purification of the particles after drug loading. Therefore the characterisation is not scientifically sound - the authors rely on uncontrolled evaporation of the unbound compounds. How can they be certain that all the unbound compound has been removed in this way before FT-IR? This technique has been used to look at the composition of bound drug - but the drugs within each EO mixture have different boiling points/vapour pressures. For the antimicrobial testing, this means what they are actually measuring is the activity of the free drug + whatever has been bound, which they are so far not able to adequately characterise because of the inaccuracy of this evaporative spectroscopic characterisation technique.
- The DLS results look strange. If the authors are really plotting the intensity autocorrelation function (G2(τ)), my understanding is that the results should show an exponential decay from close to 2.0 down to 1.0. Here the autocorrelation functions are very shallow (1.3 - 0.99), implying that the sample concentration was too low with high signal-to-noise. The authors should explain this result further - were multiple scattering events occurring, or perhaps absorbance at the laser wavelength etc? Or was it just very dilute?
Overall, the results and discussion section leaves too many questions unanswered and I'm not convinced by the experimental design (testing/characterising without purifying from free drug). Therefore, I cannot recommend publication in the article's current form.
Author Response
Reviewer
After the rapid additions from the authors, I am unfortunately left feeling that there are more issues with this article than before.
- Having the structures in the table is helpful. However, my comments surrounding the MIC of the antimicrobial compounds that make up each of the EO mixtures has not been adequately addressed. For example, what was the MIC of thymol, p-cymene, linalool, borneol, etc. against each of the pathogens? How does this compare to the MIC of the particles (containing a loaded mixture of these compounds) and to the MIC of the individually loaded compounds on the particles (calculated from FTIR and TGA)?
Thank you very much for your comments and suggestions. GC-MS analysis was performed on all EOs and EOs-functionalized nanosystems. MIC values for each of the major compounds identified were added from the literature.
- There was no purification of the particles after drug loading. Therefore the characterisation is not scientifically sound - the authors rely on uncontrolled evaporation of the unbound compounds. How can they be certain that all the unbound compound has been removed in this way before FT-IR? This technique has been used to look at the composition of bound drug - but the drugs within each EO mixture have different boiling points/vapour pressures. For the antimicrobial testing, this means what they are actually measuring is the activity of the free drug + whatever has been bound, which they are so far not able to adequately characterise because of the inaccuracy of this evaporative spectroscopic characterisation technique.
There was no sample purification after drug loading since we aimed to determine the antimicrobial potential of iron oxide-silica core-shell nanoparticles functionalized with essential oils as mixtures, and not of the isolated compounds found within the essential oils used. In this context, we believe that the GC-MS results could answer the questions asked previously.
- The DLS results look strange. If the authors are really plotting the intensity autocorrelation function (G2(τ)), my understanding is that the results should show an exponential decay from close to 2.0 down to 1.0. Here the autocorrelation functions are very shallow (1.3 - 0.99), implying that the sample concentration was too low with high signal-to-noise. The authors should explain this result further - were multiple scattering events occurring, or perhaps absorbance at the laser wavelength etc? Or was it just very dilute?
The explanations were added.
Overall, the results and discussion section leaves too many questions unanswered and I'm not convinced by the experimental design (testing/characterising without purifying from free drug). Therefore, I cannot recommend publication in the article's current form.
We hope that we addressed the issues raised by the reviewer.